# Research on a Super-Resolution and Low-Complexity Positioning Algorithm Using FMCW Radar Based on OMP and FFT in 2D Driving Scene

**DOI:** 10.3390/s23094531

**Published:** 2023-05-06

**Authors:** Yiran Guo, Qiang Shen, Zilong Deng, Shouyi Zhang

**Affiliations:** 1School of Mechatronical Engineering, Beijing Institute of Technology, Beijing 100081, China; 2Beijing Institute of Technology Chongqing Innovation Center, Chongqing 401120, China

**Keywords:** FMCW radar, OMP, FFT, particle swarm algorithm(PSO), 2D trajectory estimation, super-resolution, low-complexity

## Abstract

Multitarget positioning technology, such as FMCW millimeter-wave radar, has broad application prospects in autonomous driving and related mobile scenarios. However, it is difficult for existing correlation algorithms to balance high resolution and low complexity, and it is also difficult to ensure the robustness of the positioning algorithm using an aging antenna. This paper proposes a super-resolution and low-complexity positioning algorithm based on the orthogonal matching pursuit algorithm that can achieve more accurate distance and angle estimation for multiple objects in a low-SNR environment. The algorithm proposed in this paper improves the resolving power by two and one orders of magnitude, respectively, compared to the classical FFT and MUSIC algorithms in the same signal-to-noise environment, and the complexity of the algorithm can be reduced by about 25–30%, with the same resolving power as the OMP algorithm. Based on the positioning algorithm proposed in our paper, we use the PSO algorithm to optimize the arrangement of an aging antenna array so that its angle estimation accuracy is equivalent to that observed when the antenna is intact, improving the positioning algorithm’s robustness. This paper also further realizes the use of the proposed algorithm and a single-frame intermediate frequency signal to estimate the position angle information of the object and obtain its motion trajectory and velocity, verifying the proposed algorithm’s estimation ability when it comes to these qualities in a moving scene. Furthermore, this paper designs and carries out simulations and experiments. The experimental results verify that the positioning algorithm proposed in this paper can achieve accuracy, robustness, and real-time performance in autonomous driving scenarios.

## 1. Introduction

With the continuous improvement of technology and the reduction of industrial manufacturing costs, widespread uses of FMCW millimeter-wave radar for unmanned peripheral environment perception, positioning, obstacle avoidance, and other applications have rapidly emerged. In the fields of intelligent assisted driving or autonomous driving, the use of millimeter-wave radar signals to accurately detect the position and speed of neighboring vehicles and pedestrians in order to control vehicle speed and steering will greatly improve the safety of automatic driving. However, the positioning algorithm applied in this scenario faces the demands of high resolution and low complexity.

Currently, the FMCW radar positioning algorithm can be roughly divided into three categories: the baseline algorithm, spatial spectrum estimation methods, and the compressed sensing algorithm. The baseline algorithm uses an FFT ranging angle measurement after adding a Hamming window, which is low in complexity and simple in implementation, but exhibits low resolution. Spatial spectrum estimation methods, such as the MUSIC algorithm, use spatial smoothing filtering and feature subspace decomposition to separate the signal from the noise space [1], but they also include problems of resolution degradation and susceptibility to noise interference. The compressed sensing algorithm uses the sparseness of the objects detected to improve angle resolution, but its complexity is high [1], and it is challenging to create a complete Fourier dictionary in the scene, which directly affects the algorithm’s accuracy and operation speed.

Several researchers have conducted studies based on the FMCW localization method. For example, Zhiyi Duan et al. proposed a waveform-based FMCW algorithm for multitarget detection in 2017 [2], which completed the algorithm using FFT waveforms, but did not solve the inaccurate localization caused by beat frequency overlap. Based on this research, Yuki Tachibana et al. developed a power-based multitarget detection FMCW algorithm [3], which achieves low-complexity localization in the distance dimension by detecting the power of the reflected echo of the target and comparing it with the echo power of the same object. However, using only the power information could lose the relative angle information of the object, and the real-time angle and motion trajectory of the object would not be obtained.

In 2020, Shunqiao Sun and Athina P. Petropulu used matrix completion to compute the corresponding virtual uniform linear array (ULA) before estimating the target angle [4,5,6]. However, they verified the effectiveness of the ULA using only FFT and MUSIC. Lukas Piotrowsky et al. used an 80 GHz broadband FMCW radar sensor to measure a movable radar target at a maximum measurement range of 5.2 m and estimated the distance to the target using an algorithm that evaluates the phases [5].

These studies demonstrate that it is feasible to use FMCW radar-transmitting linear frequency modulation (FM) signals for ranging and localization. However, these studies did not combine high-resolution and low-complexity requirements, and there are limitations in their applications. Moreover, there are also radars that use other algorithms, such as the phase shift method. The frequency modulation continuous wave (FMCW) method and the phase shift method are both commonly used in radar systems for various applications. the phase shift method relies on measuring the phase shift between the transmitted and received signals. This method is less complex than FMCW and requires less bandwidth, but it has lower accuracy and is more sensitive to noise. It is commonly used for short-range applications, such as in proximity sensors and level gauges, while FMCW provides high accuracy measurements of distance and velocity and is commonly used for applications such as automotive collision avoidance and atmospheric remote sensing. Therefore, FMCW is preferred for long-range and high-accuracy applications.

This paper focuses on typical application scenarios, such as indoor object positioning and outdoor autonomous driving, simplifying them into flat two-dimensional scenarios. The first part of this paper introduces the modeling of a super-resolution positioning algorithm based on orthogonal matching tracking, including algorithm models based on FMCW multitarget positioning, object motion trajectory generation, and antenna array arrangement optimization, to improve the positioning accuracy of an aging antenna. The second part of the paper introduces the simulation experiment settings and provides an analysis of the experimental results corresponding to each algorithm. Figure 1 shows the block diagram of the FMCW radar-based multitarget localization algorithm conceived in this paper.

## 2. Modeling of Super-Resolution Localization Algorithm Based on OMP Sparse Decomposition

### 2.1. Localization Algorithm Based on OMP Sparse Decomposition

The utilization of the common OMP or MP algorithm necessitates knowledge of the sparsity of the input signal sequence, which refers to the number of signal sources present. However, in most practical scenarios of autonomous driving, the sparsity of the echo signal A is usually unknown, making it difficult to determine the number of obstacles or people beforehand. This paper proposes an algorithm to improve the traditional OMP algorithm by processing the echo signal using the FFT baseline algorithm, establishing a complete Fourier dictionary at the maximum peak value of the echo signal amplitude strength obtained by FFT. Additionally, a sparser dictionary is created at the other peaks of the FFT waveform, and OMP is utilized to sparsely decompose the overall echo signal. The estimation accuracy is determined by comparing the acceptable value of the signal residuals and the signal estimation error. This processing method is suitable for situations where the number of sources is unknown.

Therefore, to implement this algorithm, the echo signals received by the RX antenna of the FMCW radar must first undergo FFT processing, and the results can serve as a reference for establishing a low-complexity dictionary in real time. Establishing the angular dimension dictionary is simplified because the angular resolution of the radar is not uniform in the center region of the array, nor at the boundaries on either side of the angular field of view, owing to the different positions of the antenna array elements. The redundant dictionary based on the angular resolution of the antenna array is established in the highest peak of the signal waveform after FFT processing. However, on both sides of this peak, the angular resolution of the antenna array gradually decreases, as neither ∆ϕ nor sinθ in Equation (10) are linearly related. Therefore, the corresponding dictionary at the side flap waveform is not required to be constructed as a complex dictionary adapted to the best angular resolution. In Figure 2, we depict the flow of the algorithm to build the dictionary.

The cyclic flow shown in Figure 3 can solve the process of building a complete dictionary.

Here, θ is the angular resolution of the antenna array, k is a factor and is less than 1, AntDis represents the antenna array element spacing and is equal to *L* in Equation (10), and Na is the number of antenna array elements. The step size of the angular sequence corresponding to the wave crest of the FFT waveform is set to kθ, and the corresponding overcomplete dictionary is established. Similarly, the idea of establishing the dictionary at the side flaps on both sides of the highest wave peak is based on the same principle as that of establishing the overcomplete dictionary. However, the angular resolution decreases near the boundaries of both sides of the angular field of view, which necessitates a larger step size of the angular variation corresponding to the para-flap area to reduce the number of atoms in the dictionary, as well as its complexity.

Concerning the dictionary establishment of the distance dimension, its resolution has no or negligible variability regarding the reception of the antenna array elements at different positions in the radar field of view. Therefore, simplifying the distance dimension dictionary may cause the matching tracking algorithm to fail, affecting the distance estimation accuracy. We first construct the dictionary of the distance domain following the cyclic process shown in Figure 2. To calculate the distance resolution of the signal, we consider S as the slope of a linearly increasing FM signal, T as the period of the chirp signal, and B as the bandwidth of the signal; then S=B/T. The frequency resolution of a signal of length time T is $f_{res}=1/T$. Subsequently, the frequency resolution of the IF signal can be determined using Equation (1).
(1)Disres=c2S·fres=c2ST=c2B,

Setting the number of snapshots as N, we can obtain the formula for establishing the distance domain dictionary, as shown in Equation (2).
(2)DisDic(:,i)=ei·2π·(0:N−1)·S·T·(Disresc),

OMP can be utilized to perform the DOA and distance estimation of the echo signal once the dictionary is established [7,8,9]. The sparse representation coefficient of the signal is denoted as $\hat{\theta }$, and the N − 1 dimensional residual is $r_{k}=y-A_{k}{\hat{\theta }}_{k}$. In this paper, the maximum detection distance of the FMCW radar is 15 m. The sparsity of the signal is judged by satisfying $\left|r_{k}-r_{k-1}\right|<\left|r_{k}\right|*Thr$, and the corresponding sparse signal in the dictionary is the output. Otherwise, the iterative process of OMP continues, where Thr is the threshold value. To ensure the detection accuracy for the object to be detected, the Thr value of the improved OMP-based algorithm proposed in this paper is set to 2. Based on our practical testing experience, if the residual of the current iteration round is less than 1/3 of the residual of the previous iteration, we can assume that there is a jump in the matching degree in this iteration, and the dictionary atom matching the echo signal is determined.

Therefore, the main body of the OMP-based super-resolution low-complexity localization algorithm in this paper can be obtained as shown in Algorithm A1, which is attached in the Appendix A.

It is noteworthy that this algorithm does not engage in signal reconstruction. This is because its sole objective is to locate the echo signal of the object to be detected, which is akin to extracting the column in which the object’s reflected signal is present in a redundant dictionary. The decomposed signal under that index contains the information to estimate the distance and angle. This approach significantly reduces the complexity of the algorithm compared to that of compressed sensing algorithms.

### 2.2. FMCW-Based Multi-Target Localization Principle and Model Construction

As shown in Figure 4, the FMCW radar antenna relies on receiving echoes and performing signal processing to obtain information. In this paper, the signal used in the FMCW radar is a linear FM pulse chirp signal, and within a chirp period T, the frequency of the radar-transmitted signal $f\left(t\right)$ satisfies $f\left(t\right)=f_{0}+\gamma t,0\leq t<T$, and the transmitted wave signal is shown in Equation (3) [1]. The waveform of the TX (*transmit*) antenna transmitting signal is shown in Figure 5a.
(3)$$s_{TX}\left(t\right)=A_{TX}exp(j2\pi \left(f_{0}t+\frac{\gamma }{2}t^{2}\right)),0\leq t<T,$$
where $A_{TX}$ denotes the signal transmitting power, $f_{0}$ denotes the carrier frequency, γ denotes the FM slope, *T* denotes the chirp period, and *j* denotes the imaginary unit [1,10]. The FM bandwidth of the linear FM signal used in this paper is 2.5 GHz. The signal from an object, reflected by the radar at a distance of $R_{0},$ is shown in Equation (4). We assume the presence of multiple objects within the radar detection range, and the RX (*receive*) antenna picks up their signals. The echo signal waveforms of the *mth* object and the *(m + 1)th* object detected by the radar are shown in Figure 5b.
(4)$$s_{RX}\left(t\right)=A_{RX}exp(j2\pi \left(f_{0}\left(t-\tau \right)+\frac{\gamma }{2}{\left(t-\tau \right)}^{2}\right)),\tau \leq t<T,$$
where $A_{RX}$ denotes the signal reception power, $\tau =2R_{0}/c$ is the reception delay, and *c* is the speed of light. The transmit signal and the receive signal overlap on [*τ, T*], and the two signals are fed into the mixer to obtain the intermediate frequency (IF) signal, which is shown in Equation (5) [1,8]. The mixed signal waveform obtained from the RX antenna-received signal in Figure 5b and the TX antenna-transmitted signal in Figure 5a are shown in Figure 5c.
(5)$$s_{IF}\left(t\right)=s_{TX}\left(t\right)s_{RX}^{*}\left(t\right)=A_{TX}A_{RX}exp\left(j2\pi (\gamma \tau t+f_{0}\tau )\right),\tau <t\leq T,$$
where the term $\tau^{2}$ (*τ* ≪ 1) is neglected. Since the frequency of the IF signal is $f_{IF}=\gamma \tau$, R0=(c2γ)fIF.

This paper lists the antenna array as a common linear array. In the horizontal plane, the $N_{a}$ equivalent virtual antenna array with aperture L is uniformly arranged. We establish a coordinate system with the center of the antenna array as the origin, the antenna array as the *x*-axis, and the area perpendicular to the antenna array as the *y*-axis [1,5,7,11,12,13], as shown in Figure 6a,b.

Supposing that the distance from object $O_{k}$ to the center is $r_{k}$ and the angle with the *y*-axis is $\theta_{k}$, then the corresponding coordinates are shown in Equation (6):(6)$$X_{k}=r_{k}sin\theta_{k},Y_{k}=r_{k}cos\theta_{k},$$

The two-way echo distance of the *n*th antenna (with coordinates {$x_{n}=-L/2+nL/N_{a}-1,y_{n}=0$}, *n* = 0, 1, ⋯, $N_{a}-1$.) is shown in Equation (7):(7)$$R_{n,k}=2\sqrt{{\left(x_{n}-X_{k}\right)}^{2}+{\left(y_{n}-Y_{k}\right)}^{2}},$$

For such an echo distance of an object, the FMCW radar at the moment *t* and the received IF signal on the *n*th antenna are shown in Equation (8) (uniform sampling will be carried out hereafter, and the formula $T_{s}t$ is simplified to *t.*):(8)sn,kt=akej2πγTstτ+2πf0τ=akej2πγTstRn,k/c+2πf0Rn,k/c,t=0,1,2,⋯,[TTs]−1,
where $a_{k}$ denotes the reflectivity of the object, $T_{s}$ is the sampling interval, c is the speed of light, and [$T/T_{s}$] denotes rounding. At moment t in a chirp period, the IF signal received by the nth antenna is a mixture of K target objects with noisy IF signals [1], as shown in Equation (9):(9)$$z_{n}\left(t\right)=\sum_{k=1}^{K}s_{n,k}\left(t\right)+w_{n}(t),\;\;\;t=0,1,2,\cdots ,[T/T_{s}]-1,$$
where $w_{n}$ means noise.

In the design of traditional algorithms based on OMP, it is often necessary to create a redundant dictionary that builds up the dimensions of distance or angle. If an overcomplete dictionary of position and angle information is established at the same time, it will add unnecessary complexity to the algorithm. To reduce this complexity, this paper uses the following process for the establishment of sparse dictionaries.

The step size of the angular search is determined based on the angular resolution of the FMCW radar. The difference in distance between the object and the two antennas causes a phase change in the FFT peak, and the AOA is estimated by the phase change. The phase change is mathematically established with Equation (10) [9,14]:(10)∆ϕ=2πΔdλ,

In the two-dimensional plane, the geometric relationship is Δd=Lsinθ, where Δd is the difference in the distance of the echo signal received between the RX receiving antennas, and *L* is the distance between the two antenna elements. The angle between the line of the object and the origin of the coordinate system and the *y*-axis can be found with Equation (11):(11)θ=sin−1(λ∆ϕ2πL),

According to Equation (11), ∆ϕ depends on sinθ, and they are not linearly related. When *θ* is tiny, sinθ can be approximately equal to *θ*. Therefore, θ is more accurately estimated when θ is close to 0°, and the accuracy decreases when θ tends to 90°. It is supposed that there is only one TX antenna and one RX antenna in the radar, and that the distances between them are the same as the antenna-spacing parameter in the experimental section of this paper. The FMCW radar used in this paper has a maximum angular field of view of ±50°, and the resulting estimation error at the boundary is about 6°. This error is acceptable in daily application scenarios. We set the step of angle variation to 0.1° in the range of ±25°, and increase the step to 0.2° in 25°~50° and −25°~−50° to reduce the error.

Then, from Equation (8), we can establish a dictionary of echo signals that the center of the antenna array may receive from objects at different distances. Using this dictionary, the actual signal received by the radar can be analyzed and matched, and sparse processing can be performed to obtain the angle and position information of the target object [3,12]. This is also the method used by traditional OMP algorithms to build a complete dictionary.

## 3. Particle Swarm Optimization Algorithm for Improving the Robustness of a Super-Resolution Localization Algorithm When the Antenna Is Aging

Suppose an antenna array element is out of order. In such a scenario, the estimation of the distance domain remains unaffected, but the estimation of the angle domain may be negatively impacted. This is exemplified in Figure 7. However, if the antenna array layout is considered an unknown variable, and its location is rearranged using the optimization algorithm, the resulting layout is a virtual array. This can be deemed optimal if it minimizes the residuals estimated by the algorithm proposed in Section 2.2.

The estimation of the distance domain is not affected by the aging antenna array, but the angular resolution at the aging antenna is affected. We use the sparse decomposition of the distance domain of the echo signal using the OMP algorithm mentioned in Section 2.1 to obtain the corresponding sparse expression Sest in the distance domain and then find the dictionary index Index corresponding to the peak representation in the sparse signal. The pseudoinverse of ${Dis}_{Dic}\left(:,Index\right)$ is found and multiplied with the echo signal to obtain the virtual echo signal with a constant distance domain result, recorded as ${Data}_{tmp}$. When an antenna array element loses its role because of aging, if the remaining intact element is rearranged uniformly, the antenna array element spacing is $L/(N_{a}-1)$, and the uniform array is recorded as ${Ant}_{loc}$.

Then, the angle domain dictionary ${AngGen}_{Dic}(:,i)$=$e^{i\cdot 2\pi \cdot {Ant}_{loc}/\lambda \cdot \sin(Ang(i))}$ is established for the virtual echo signal ${Data}_{tmp}$. Based on this dictionary, the virtual echo signal ${Data}_{tmp}$ is sparsely decomposed using the OMP-based super-resolution algorithm proposed in this paper, and the corresponding residuals are calculated. The residual maximum is the initial population of the particle swarm optimization algorithm [13,15], and the object of optimization is the distance correction value between each antenna element. If we set the position of the antenna elements as ${Ant}_{Loc}$, Dic() is the compressed dictionary construction function, and ${omp}_{res}$() represents the residual function to calculate OMP, then there is an objective function, as shown in Equation (12).
(12)$${Ant}_{Loc}={argmin}_{AntLoc}(\max(omp_{res}\left(Dic\left(AntLoc\right),\;\left[Z_{1}\left(f_{1}\right),Z_{2}\left(f_{1}\right),\cdots ,Z_{Na}\left(f_{1}\right)\right]\right),\cdots ,omp_{res}\left(Dic\left(AntLoc\right),\;\left[Z_{1}\left(f_{k}\right),Z_{2}\left(f_{k}\right),\cdots ,Z_{Na}\left(f_{k}\right)\right]\right),$$

In this paper, we chose the particle swarm algorithm (PSO) to solve this objective function. By updating the iterations of the particle swarm, the minimum residual function is calculated under different arrangements, and the correction value of the best antenna array element spacing ${Ant}_{cor}(best)$ is obtained for a given iteration round. The final arrangement of the antenna array elements is ${Ant}_{Loc}+{Ant}_{cor}(best)$, which completes the solution of the actual antenna array placement under the condition of the highest angle estimation accuracy.

The pseudocode of the antenna array element position rearrangement algorithm based on PSO is given next, as shown in Algorithm A1, which is attached in the Appendix A. In Section 6.2, we carry out simulation experiments to verify whether the accuracy of the positioning of the object being detected is corresponds to the intact antenna array.

## 4. Super-Resolution Low Complexity Localization and Object Motion Trajectory Estimation Based on Single-Frame Signals

In Section 3 of the algorithm modeling process, we aim to enhance the robustness of the proposed algorithm in Section 2.1 To achieve this, we employ a fine-tuned optimization method based on a uniformly redistributed array to obtain the optimal virtual antenna array element arrangement. However, the computational process of the proposed algorithm in Section 2.1 and Section 3 is solely for one chirp period in a frame of the IF signal; if all the chirp periods in such a frame are needed to build the corresponding dictionary, then the algorithm’s complexity for solving one frame will be multiplied by the number of chirp periods. This increase in complexity can make it challenging to meet the low-complexity requirements in online moving scenarios.

Therefore, to achieve distance, angle, and velocity estimations in actual moving scenarios, we need to further simplify the modeling process of the algorithm. First, we extract a frame of the echo IF signal received by the FMCW radar. Figure 8 illustrates the process of obtaining a 3D data matrix of IF signals in the time, antenna, and chirp domains. The K beat signals of length *n* are connected in series to form an *n* × K matrix. We denote $Y_{l}$ by the two-dimensional data matrix at the *lth* chirp. Then, we obtain the three-dimensional data matrix of the radar-received IF signal, as shown in Figure 9.

In order to effectively compress all periodic signals in the distance domain (angle domain), the construction of a comprehensive dictionary of reflected signals at various distances and angles is imperative. The principle of building the dictionary is akin to that presented in Equation (9) and Figure 3 of the previous algorithm, which was based on orthogonal matching tracking. In order to minimize the size of the compressed dictionary while maintaining a remarkable level of accuracy, we establish the dictionaries of distances and angles as follows:First, a compressed perceptual dictionary Dic(Dis) (or Dic(Ang)) of the reflected signals at different distances (angles) is constructed.If the signal analyzed is the first cycle of a frame, then the dictionary should include all distance or angle domain signals; otherwise, it means that the signal analyzed is a different cycle of a frame, and only the dictionary needs to be built in the vicinity of the previous distance estimation result.

The overall low-complexity dictionary building process in the angle and distance domains is illustrated in Figure 10. It is noteworthy that the bandwidth of the chirp signal used in the simulation experiments in this paper is approximately 2.5 GHz, and the distance resolution ${Dis}_{res}=3\;cm$ of the FMCW radar used in this paper can be determined from Equation (11). Therefore, the minimum distance between interpolations in the distance domain data correlation should be no less than 3 cm. The angular resolution near the center of the array is utilized instead of that under the whole angular field of view, according to Equation (11).

The distance and angle domain dictionaries of the signals in all the cycles in a frame of the IF signal, except for the first cycle, can be obtained from the algorithm shown in Figure 10. The angle and distance information of each object in a frame of the IF signal can be obtained by estimating the angle and distance of the signal in each cycle using the algorithm proposed in Section 2.1. Consequently, the object can be localized.

For the velocity discrimination problem of multiple objects in moving scenes, this paper considers two implementation ideas. One is the more common velocity-dimension DFT algorithm, which posits that the overall distance–velocity spectrum of multiple objects is a linear combination of the spectra of individual reflected objects because the DFT (*discrete Fourier transform*) is a linear transformation, so objects with different distances and velocities can be detected from the peaks in the distance–velocity spectrum. Let the angular frequency resolution $w_{res}=2\pi /N_{c}$ for a discrete time signal of time length $N_{c}$ [16,17,18,19]; the velocity resolution $v_{res}$ is shown in Equation (13).
(13)vres=λ4πTc·wres=λ2NcTc=λ2TF,
where $T_{F}$ is the frame duration, and λ is the wavelength. Another idea is to use the algorithm proposed in this paper to calculate the Euclidean distance between the positions of the object at different cycles within one frame of the signal by using the localization results of the target object at different cycles, and from this, the average velocity of each cycle is obtained as the instantaneous velocity of the object (because the period of a single chirp signal in one frame of the IF signal is very short).

## 5. Complexity and Resolution of the Proposed Algorithm in This Paper

In Section 2, Section 3, and Section 4, we completed the introduction of the algorithms proposed in this paper. Here, we calculate the complexity of the FFT- and OMP-based super-resolution low-complexity localization algorithms, namely the classical FFT algorithm, MUSIC, and OMP.

We assume that the number of elements in the antenna array is $N_{a}$, the number of beat frequencies of the chirp signal is *N,* and the number of DFT points is $N_{fft}$. Then, we can obtain the complexity of FFT as O($N_{a}$*N* · log(*N*) + $N_{a}N_{fft}\;\cdot \;$log($N_{fft}$)).

We assume that the length of the vector theta formed by the angles traversed in the MUSIC algorithm is *L*. That is, theta satisfies the equation:(14)$$theta=\left[-\theta_{max}:(2\cdot \theta_{max}/L):\theta_{max}\right],$$
where $\theta_{max}$ is the angle at the maximum boundary of the radar’s angular field of view. Since the covariance matrix is calculated in the MUSIC algorithm, as well as in the eigendecomposition process, such matrix multiplication is calculated throughout the cycle for each angle value, so the complexity of MUSIC is O($N^{3}\cdot L$).

Let us consider the complexity of the classical OMP algorithm. We assume that the dimensionality of the angle dictionary created is the same as the length of the angle vector in MUSIC, i.e., *L*. After this operation, the angular resolution of the classical OMP algorithm and MUSIC will theoretically be at the same level. We set the number of objects detected as M. Then, the complexity of classical OMP can be obtained as O($M\cdot L\cdot N_{a}$).

Regarding the complexity of the super-resolution low-complexity localization algorithm proposed in this paper, we can calculate the complexity of the part of the algorithm that analyzes the FFT waveform of the echo signal and builds a dictionary adapted to it is O(*N*·log(*N*) + $N_{a}$*N* · log(*N*) + $N_{a}N_{fft}\cdot log$($N_{fft}$)). We set the dimension of the reduced-complexity dictionary matrix as $L_{low}$, and the number of objects detected is set to M. Then, we can obtain the complexity of the super-resolution low-complexity localization algorithm proposed in this paper as O(*N*·log(*N*) + $N_{a}$*N* · log(*N*) + $N_{a}N_{fft}\cdot log$($N_{fft}$)) + O($M\cdot L_{low}\cdot N_{a}$).

In order to compare the complexity of the algorithms, we set the number of detected objects M to be 2, the number of vector elements of the angle *L* to be 1000, the number of DFT points $N_{fft}$ to be 256, and the number of arrays in the antenna array $N_{a}$ to be 4. If the number of beat frequencies *N* of the chirp signal is taken as the independent variable and the algorithm complexity as the dependent variable, then we can obtain the complexity curve of the above algorithm, as shown in Figure 11. It can be clearly seen that the algorithmic complexity of MUSIC increases significantly with the number of beat frequencies N of the signals used. Conventional FFT has the lowest complexity. Due to the low dimensionality of the matrix constructed, the algorithm proposed in this paper is less complex than that of the classical OMP.

Next, we discuss the resolution that can be achieved by the positioning algorithms proposed above. Distance resolution is the ability to discriminate between two or more objects at different distances. Fourier transform theory states that the observation window (length T) can resolve frequency components at intervals greater than 1/THz. This means that two IF signals can be distinguished, if the frequency difference satisfies the relationship given in Equation (15).
(15)∆f>1T,

If the linear FM slope of the chirp signal emitted by the antenna is set as γ, and the distance resolution is represented by $d_{res}$, then we can obtain the relation equation from equation: ∆d>c/2γ·T=c/2B. Distance resolution $d_{res}$ depends only on the bandwidth of the linear FM pulse sweep B; that is, $d_{res}=c/2B$. Then, there is not much difference in distance resolution between the above four algorithms. Next, we focus on the angular resolution, which is low for the conventional FFT algorithm, and if we express it as ${Angle}_{res}(FFT)$, it satisfies the equation ${Angle}_{res}(FFT)=2\pi /N_{a}$. In fact, the experiments we performed in Section 6.1 show that the angular resolution of FFT is indeed very poor. MUSIC requires an estimate of the covariance matrix to be constructed from the received signal vector, which affects the accuracy of the algorithm in estimating the angle. Therefore, the number of elements in the array, the SNR of the received signal, and even the step size of the traversal angle of the arrival process all affect the angular resolution of the MUSIC algorithm ${Angle}_{res}(MUSIC)$. The angular resolutions of traditional OMP and the algorithm proposed in this paper are mainly influenced by the angular measurement accuracy of the radar, the signal-to-noise ratio, and the accuracy of the dictionary construction, as they all make use of the sparse characteristics of the signal. The angular resolution of the proposed algorithm is essentially comparable to that of traditional OMP, and is improved by the fact that it can build a more precise dictionary near the center of the angular field of view. In the experiments in Section 6.1, we will give the values of the angular resolution that can be achieved by the different algorithms.

## 6. Simulation Experiments and Results

In each scenario, it is assumed that *K* (to be determined) is an object within the detection range of the radar [9,13,14,16,19,20] (a sector with a radius of 15 m from the origin and an angle of 100° from the center of the circle).

### 6.1. Localization Simulation Experiments Based on OMP Sparse Decomposition

The distances of the object to be detected from the origin of the antenna coordinates were set to 10.1 and 11.5 m, respectively, and the angles of the line regarding the origin and the *y*-axis were set to $1.4^{{}^{\circ}}$ and ${-1.7}^{{}^{\circ}}$. After modulating the waveform of the electromagnetic wave, $N_{TX}$ transmitting antennas will transmit in turn, and $N_{RX}$ receiving antennas will receive the returned signal in turn; because this period is very short, it can be equated to $N_{TX}\times N_{RX}$ antennas transmitting and receiving at the same time. All simulations and experiments in this paper used the FM signal and the relevant parameters of the antenna, as shown in Table 1.

The signals collected by the 12 equivalent virtual antenna arrays contained the echo signals of two objects, mixed with Gaussian white noise, and the signal-to-noise ratio was set to 30 [21,22,23,24]. We used the classical FFT algorithm, the localization algorithm based on the OMP orthogonal matching tracking proposed in this paper, and the classical 2D-MUSIC algorithm to estimate the positions and angles of two objects. The results are shown in Figure 12a–e.

As exemplified in Figure 12b, the classical MUSIC algorithm only manages to distinguish one angle, and the angle estimate obtained by the solution is 1.26°. This shows a significant error compared to the preset values of −1.7°, 1.4° in the simulation experiment. Moreover, it is also quite challenging for FFT to distinguish two objects.

As depicted in Figure 12c,d, when the number of antenna elements is increased and other parameters remain unchanged, MUSIC can exhibit wave peaks covering different objects. However, the wave peaks are not clearly distinguished, even when the number of antenna elements is increased, to a certain extent. At this point, the baseline FFT algorithm can already distinguish two objects. This observation leads us to speculate that MUSIC introduces some noise in the smoothing filtering process, leading to angle estimation errors. Furthermore, its angle resolution is not exceptional when the number of antenna array elements is small at a low signal-to-noise ratio.

However, when the signal-to-noise ratio of the signal increases, as demonstrated in Figure 12e, MUSIC can distinguish two objects more effectively, while FFT fails to distinguish two objects. It is worth noting that the specialization filtering and spatial spectrum estimation process of the classical MUSIC algorithm itself exhibits high algorithmic complexity. In contrast, the algorithm proposed in this paper can estimate the two objects remarkably well in the above cases, and the angle estimates are −1.62° and 1.43°, respectively. This is more similar to the true values of the angle settings, which further underscores the superiority of the super-resolution capability of the proposed algorithm.

We conducted the experiments for the above algorithms in the same code-running environment; the computer used was a Lenovo ThinkPad 2018, and its CPU processor was an Intel(R) Core™ i7−8550U CPU@1.80 GHz 1.99 Hz. Then, we could compare the running times of the different algorithms in terms of the angular resolution that can be achieved under different conditions, as shown in Table 2.

To gain deeper insights into the performance of the three algorithms, we conducted a comprehensive analysis of the root-mean-square error curves of angle estimation under diverse signal-to-noise conditions for signal data, with all other parameters being the same. The results of this analysis are presented in Figure 13.

It is worth noting that the baseline FFT algorithm exhibits a higher mean-square error in estimating the angle, whereas the mean-square error of MUSIC and the OMP-based algorithm proposed in this paper are both relatively low. Moreover, when the signal-to-noise ratio is low, the accuracy of MUSIC and the proposed algorithm are quite similar. However, as the signal-to-noise ratio increases, the accuracy of the proposed algorithm is better than that of MUSIC. This observation further underscores the superior stability of the proposed algorithm under noise interference conditions.

### 6.2. Simulation Experiments on Improving the Positioning Accuracy of Aging Antenna Arrays Using Particle Swarm-Based Algorithms

In this section, the localization performance of the antenna array arrangement obtained by using the particle swarm optimization algorithm is verified through numerical simulation experiments. Two objects are set to be detected by the antenna array, the angle between their lines and the center of the antenna array and the *y*-axis are −0.31° and 0.35°, respectively, and their distances from the center of the antenna array are 12.0 m and 12.2 m, respectively. Suppose one antenna array element in the original antenna array near the detected object is aged, resulting in its inability to receive echoes and estimate its own position, producing poor angular resolution, which may lead to the localization of two objects in proximity to one object. By using the localization algorithm proposed in Section 2.2 of this paper and the FFT algorithm, the aging antenna array is used to process the echoes, and the localization results are obtained, as shown in Figure 14.

Assume that the original equivalent virtual antenna array has 86 arrays with uniform linear distribution, and one of the antenna arrays fails because of aging. The arrangement of the remaining 85 antenna elements was optimized using the particle swarm algorithm, and we describe the flow of the optimization algorithm in Section 3. The optimal placement of the virtual antenna array elements was obtained, as shown in Figure 15, then compared with the estimated angles and distances obtained by positioning the non-aged antenna array elements, as shown in Figure 16. We used the proposed algorithm and FFT again on the optimized antenna virtual array for angle and distance estimation, and the results are shown in Figure 17. Based on this, five Monte Carlo simulation experiments were conducted, and five groups of angle and distance values were randomly selected, according to the set FMCW radar detection distance and the angular field-of-view size, including one group of small angles and four groups of large angles. The positioning results obtained from the virtual array are shown in Table 3, Figure 18.

As we can glean from Figure 14, Figure 15, Figure 16, Figure 17 and Figure 18 and Table 3, the issue at hand revolves around the aging antenna array, which is equivalent to missing array elements. This leads to the signal pulling up the side flaps, thus causing a false alarm phenomenon in the proposed algorithm. To rectify this issue, we employed the virtual antenna array after rearrangement by the particle swarm algorithm-based array alignment algorithm in this paper, as its positioning and resolution capabilities meet the requirements when using the positioning algorithm.

However, the estimation of the object angle shows some deviation from the original experimental setting data, with the deviation range being 0.01°~0.9°. Conversely, the deviation in the estimation of the distance dimension is very small, ranging from 0 to 0.18 m. The maximum linear deviation distance of localization is 0.0030 m, which we believe can meet the target application scenarios, such as autonomous driving. The array rearrangement algorithm proposed in this paper can increase the robustness of the OMP-based super-resolution low-complexity localization algorithm when the antenna array elements are aging, ensuring that the antenna still exhibits a comparable localization accuracy.

### 6.3. Experiment on Estimating the Trajectory and Velocity of An Object Using a Frame of the IF Signal

We set the initial distance of two moving objects (people) from the antenna as 12 m; the angle with the antenna array center line as 2° and −2°, respectively; and the initial velocity as 0 m/s. The echo signals of two moving objects detected by the FMCW radar were analyzed and the dictionary was built in the same manner shown in Section 2.2. A single frame of the chirp signal, which contains 32 cycles, was removed and processed for distance domain compression and OMP-based orthogonal matching tracking; only the distance domain-compressed images under the 8th, 16th, 26th, and 32nd cycles are shown here (Figure 19a).

The targets to be measured in our experiment were two volunteers that moved randomly within the FMCW radar detection range, according to their own will, and we did not know their trajectories in advance. The experiment used a millimeter-wave radar module with a computer host computer for data acquisition. The sensor used was an HLK-LD7903A millimeter-wave radar module, which was connected to a computer (PC) [3], and the PC used was a Lenovo ThinkPad 2018. Its CPU processor was an Intel(R) Core™ i7-8550U CPU@1.80 GHz 1.99 Hz. The experimental setup is shown in Figure 20.

As shown in Figure 19a, the two target objects were initially at almost the same distance from the antenna, after which the two objects moved away from each other, and the distances gradually pulled apart. Using the localization algorithm mentioned in this paper and the iterative algorithm in Section 4 until the signal was estimated for 32 cycles within one frame of the signal, the results of the estimation of the position and angle $Z_{n}(f)$, i.e., the corresponding position points in each cycle of the different objects, were obtained. The position and angle ranges were estimated using the position and angle resolutions, except for the first cycle, and the estimates obtained by the algorithm in this paper were used as the most probable distance and angle estimates for this cycle. The center of the antenna array was used as the origin to establish the coordinate system [1], and all the points that met the resolution requirements and were closest to the real position were connected in a line to obtain the trajectory of multiple objects, as shown in Figure 19b. The distance between the two objects in adjacent cycles was calculated, and their average velocities in the cycle were obtained. The estimated velocities were compared with the velocities of the objects at the corresponding moments in the actual simulation, as shown in Figure 21. The velocity–distance domain FFT 3D maps of the signals of the two objects in the 28th cycle of the second half of the motion are shown in Figure 22. Figure 23 shows the comparison between the results of the velocity estimation of two moving objects using the velocity-dimension FFT algorithm and their actual velocities. The difference between the velocities of the two detected targets and their actual velocities using the algorithm proposed in this paper and the velocity-dimensional FFT method was taken as the residual, as shown in Figure 24, for a comparison of the residuals of the results of the velocity estimation of the objects using the two methods with the actual velocities. We can see that the results of velocity estimation using this paper’s algorithm for object 2, with small velocity variations, are close to the accuracy of the velocity-dimensional FFT. For object 1, with more drastic velocity changes, the velocity estimation accuracy using the algorithm in this paper is better than that of the velocity-dimensional FFT algorithm, with an accuracy improvement of approximately 30%. Therefore, we believe that using the algorithm proposed in this paper can accomplish the velocity estimation of objects using one frame of the IF signal, which can meet the demand for real-time velocity and trajectory estimation.

## 7. Conclusions

In this paper, we propose a low-complexity OMP-based super-resolution localization algorithm based on the FMCW millimeter-wave radar model for the position and angle estimation of objects for two-dimensional autonomous driving scenarios. The algorithm improves the resolving power by 2 and 1 orders of magnitude, respectively, compared to the classical FFT and MUSIC algorithms in the same signal-to-noise environment, and its complexity can be reduced by about **25–30%** with the same resolving power as the OMP algorithm. Founded on the positioning algorithm proposed in this paper, we use an algorithm model based on particle swarm optimization to rearrange the aging antenna array for the problem of degradation of positioning accuracy. Using simulation experiments, we show that our proposed virtual antenna array rearrangement algorithm can provide an aging antenna array with the same positioning accuracy as that of an intact antenna. Moreover, based on the localization algorithm proposed in this paper, the position and angle of the object can be estimated from a single frame of the IF signal, and the motion trajectory and velocity of multiple objects can be obtained. In this paper, simulations and experiments are designed to verify the localization accuracy of the proposed algorithm, the feasibility of improving the robustness of the algorithm when the antenna ages, and the trajectory and velocity estimation in mobile scenarios. The experimental results show that the proposed algorithm improves the super-resolution capability and stability under a low signal-to-noise ratio compared with current widely used algorithms, such as FFT and MUSIC, for the mobile scenario of autonomous driving, in addition to meeting real-time requirements.

## Figures and Tables

**Figure 1 sensors-23-04531-f001:**
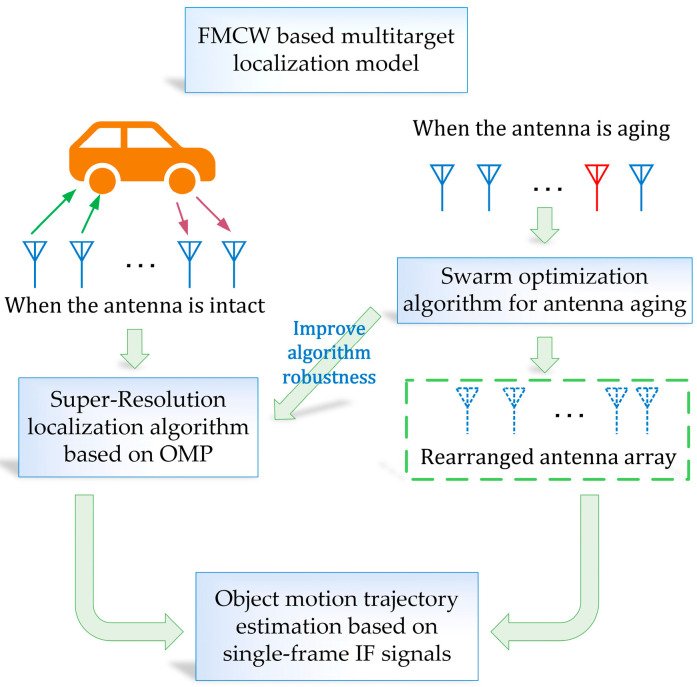
The relationship between the sections of this paper.

**Figure 2 sensors-23-04531-f002:**
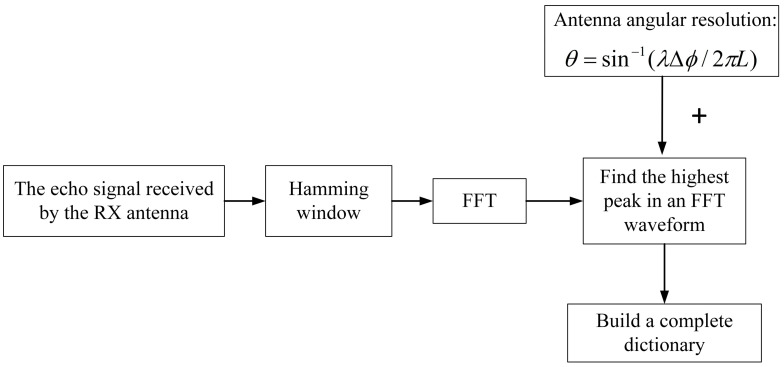
Algorithm flow for building angle-dimensional dictionaries.

**Figure 3 sensors-23-04531-f003:**
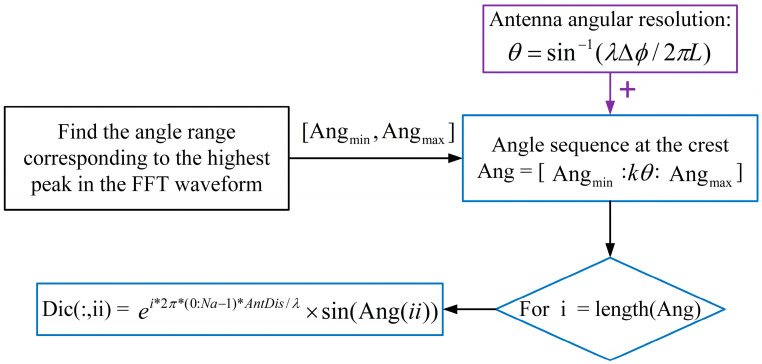
Algorithmic loop process for building a complete angle-dimensional dictionary.

**Figure 4 sensors-23-04531-f004:**
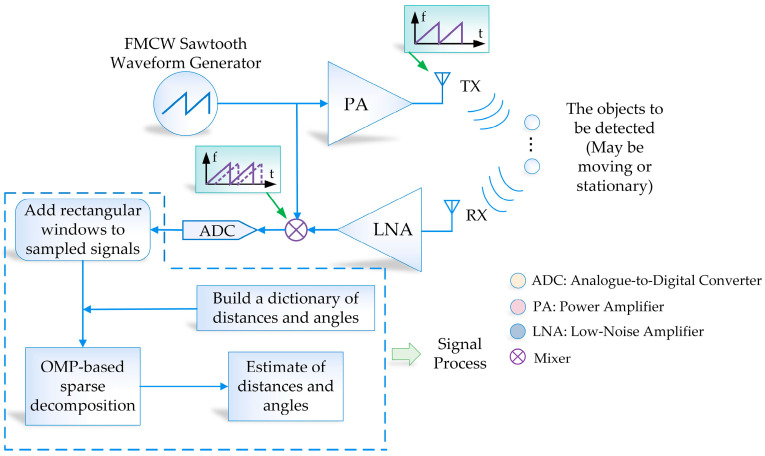
Block diagram of the principle components of FMCW multi-target localization.

**Figure 5 sensors-23-04531-f005:**
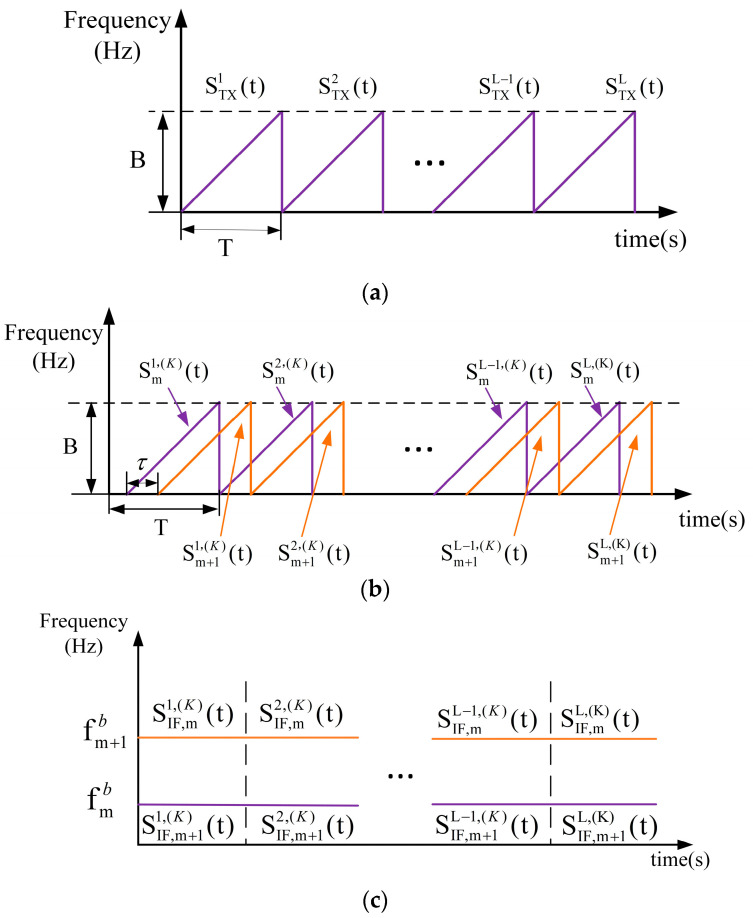
The waveform structure of the TX signal, the RX signal, and the IF signal. (**a**) Waveform diagram of the signal transmitted by the TX antenna (B represents the bandwidth of the FMCW signal). (**b**) The *mth* and (*m* + 1)*th* objects’ echo signal waveform plot of the signal received by the RX antenna (B represents the bandwidth of the FMCW signal). (**c**) Waveform diagram of the mixed signal.

**Figure 6 sensors-23-04531-f006:**
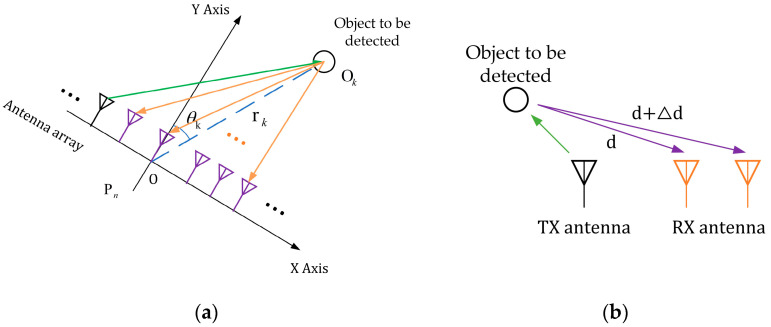
Schematic diagram of the model of the linear uniform array of antennas and the schematic diagram of the received signal of the antenna array element for angle estimation. (**a**) Antenna array layout model; (**b**) antenna array element estimation AOA schematic.

**Figure 7 sensors-23-04531-f007:**
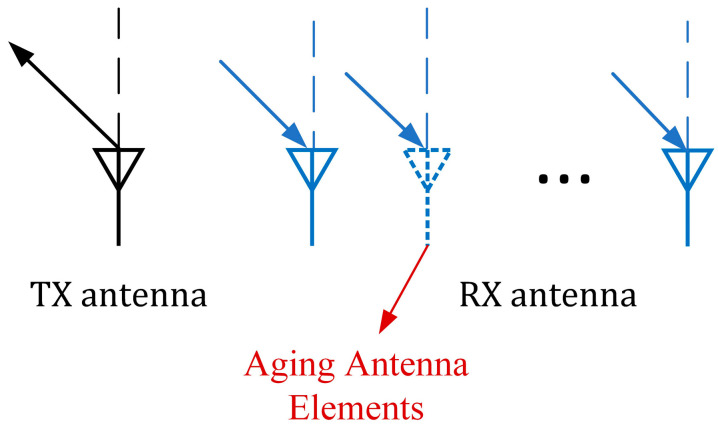
An aging antenna array element cannot receive the echo signal at the predetermined spacing, which directly affects the angular resolution.

**Figure 8 sensors-23-04531-f008:**
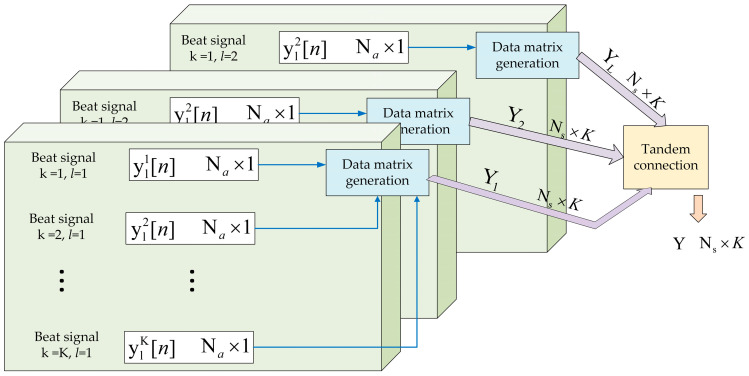
The process of generating a 3D data matrix in the time domain, antenna domain, and chirp signal domain.

**Figure 9 sensors-23-04531-f009:**
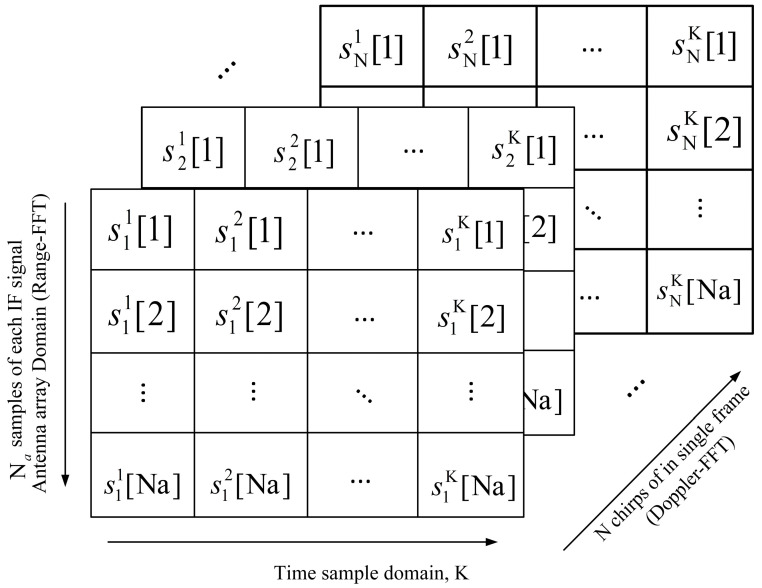
The three-dimensional representation of the radar-received IF signal.

**Figure 10 sensors-23-04531-f010:**
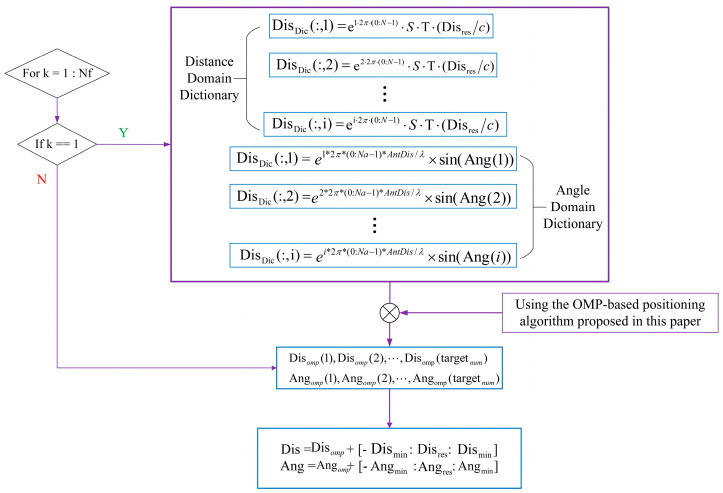
Simplified distance and angle domain dictionary building process for low complexity requirements in mobile scenarios.

**Figure 11 sensors-23-04531-f011:**
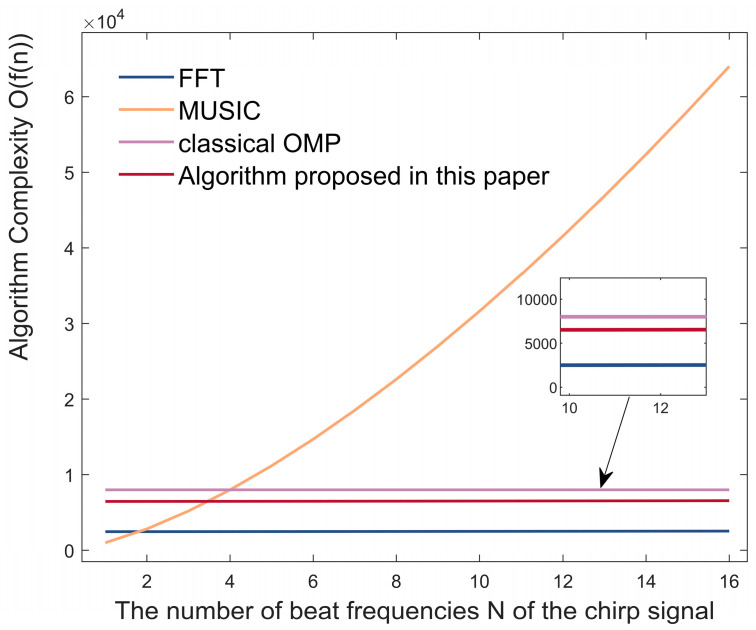
Algorithmic complexity of four localization algorithms for FMCW radar.

**Figure 12 sensors-23-04531-f012:**
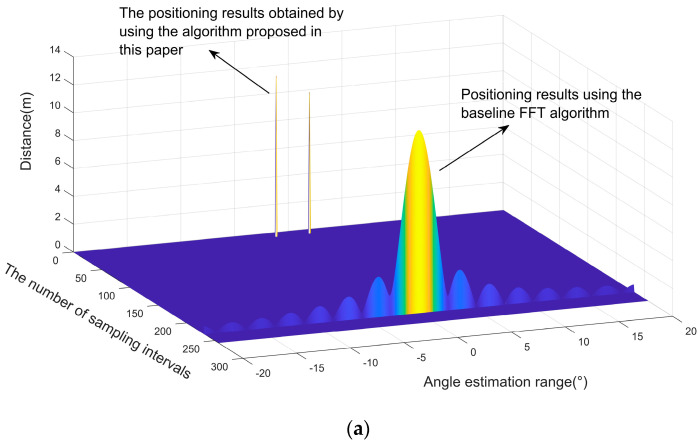
(**a**) Simulation results using the classical baseline algorithm (FFT) and the OMP-based localization algorithm proposed in this paper. (**b**) Comparison of the simulation results between the conventional FFT algorithm and the MUSIC algorithm using the orthogonal matching tracking-based localization algorithm in this paper with a 12-array uniform linear array, a signal-to-noise ratio of 30, and the number of snapshots set to 256. (**c**) Comparison of the simulation results between the conventional FFT algorithm and the MUSIC algorithm using the orthogonal matching tracking-based localization algorithm in this paper with a uniform linear array of 24 elements, a signal-to-noise ratio of 30, and the number of snapshots set to 256. (**d**) Comparison of the simulation results between the conventional FFT algorithm and the MUSIC algorithm using the orthogonal matching tracking-based localization algorithm in this paper with a uniform linear array of 48 elements, a signal-to-noise ratio of 30, and the number of snapshots set to 256. (**e**) Comparison of the simulation results between the conventional FFT algorithm and the MUSIC algorithm using the orthogonal matching tracking-based localization algorithm in this paper with a uniform linear array of 24 arrays, a signal-to-noise ratio of 50, and the number of snapshots set to 256.

**Figure 13 sensors-23-04531-f013:**
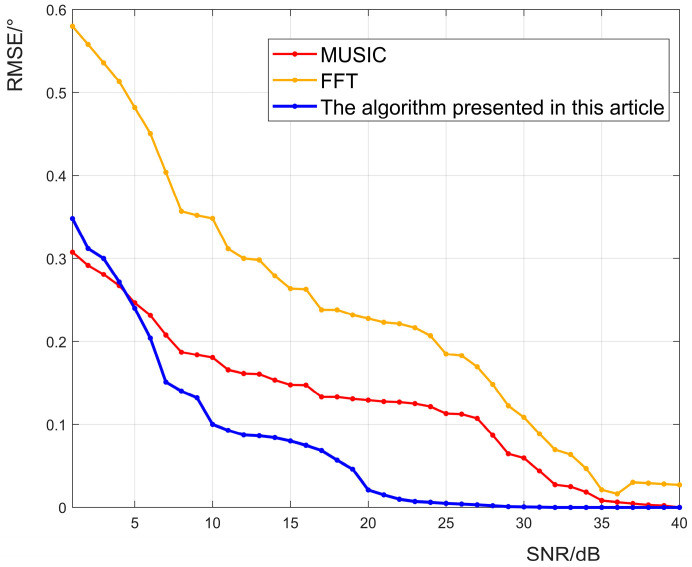
The root-mean-square error curves estimated using the proposed algorithm, the FFT algorithm, and the MUSIC algorithm angles at different signal-to-noise ratios.

**Figure 14 sensors-23-04531-f014:**
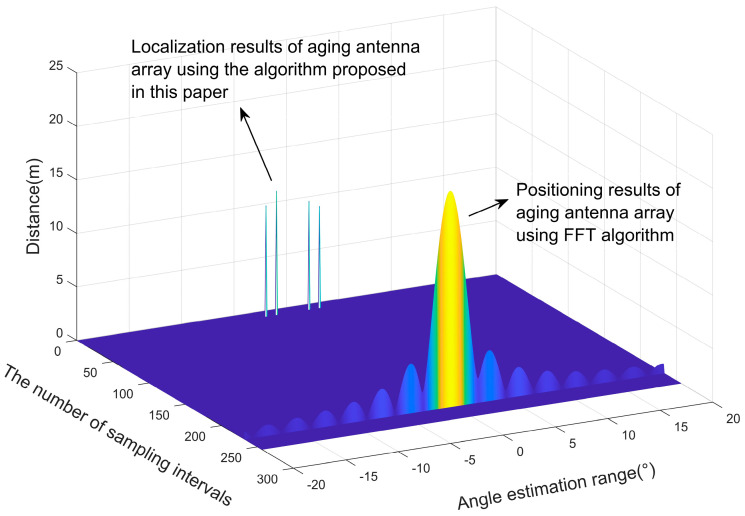
Comparison of the localization results of the aged antenna array using the proposed algorithm and the FFT algorithm.

**Figure 15 sensors-23-04531-f015:**
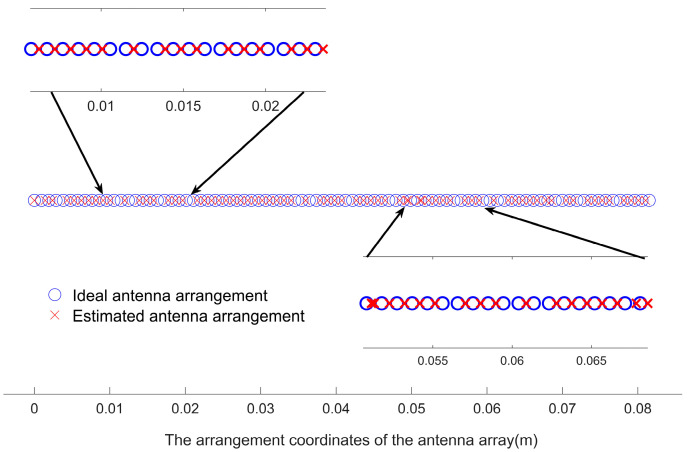
Antenna array arrangement before and after optimization using the particle swarm based array rearrangement algorithm proposed in this paper.

**Figure 16 sensors-23-04531-f016:**
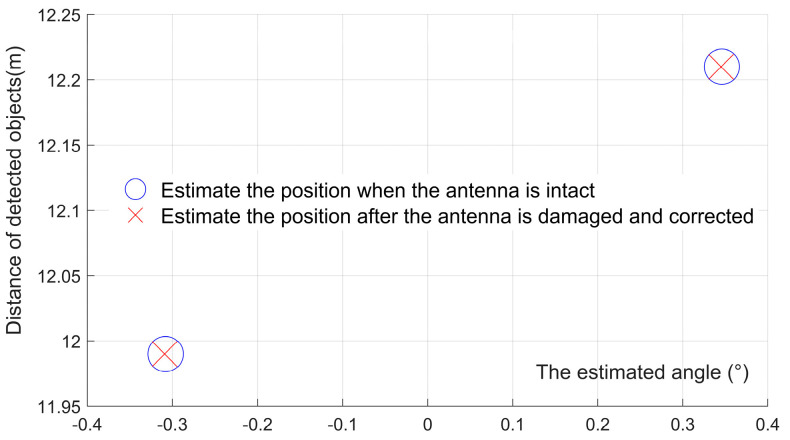
The estimation of the position and angle after the optimized arrangement, when the antenna is intact and when it is aging.

**Figure 17 sensors-23-04531-f017:**
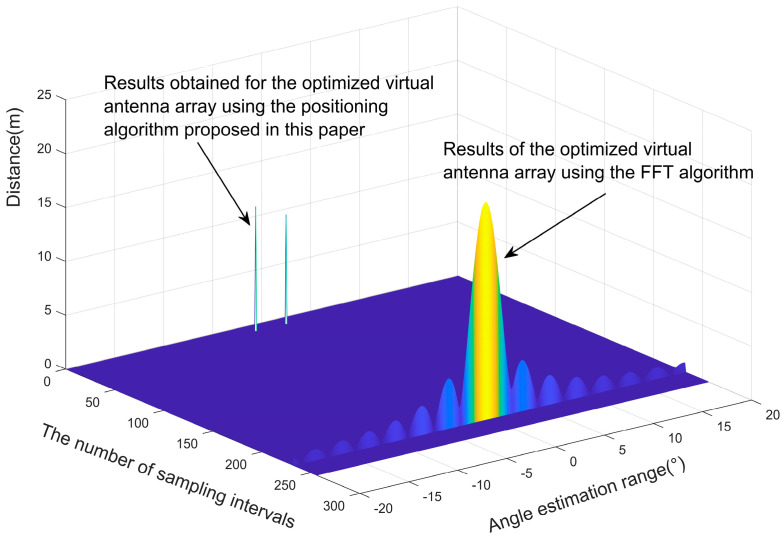
The localization results of virtual arrays obtained by optimizing the rearrangement algorithm proposed in this paper using the algorithm in this paper and the FFT algorithm.

**Figure 18 sensors-23-04531-f018:**
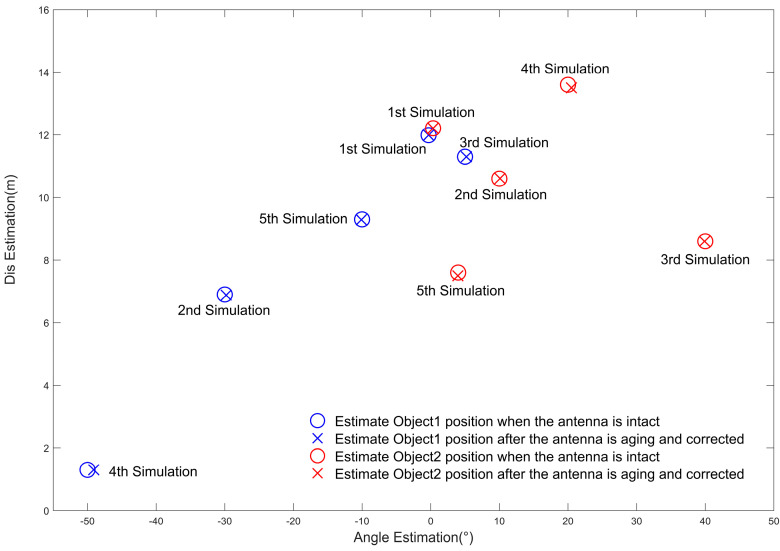
The localization results of the virtual antenna array in 5 Monte Carlo simulation experiments compared with the experimental setup data.

**Figure 19 sensors-23-04531-f019:**
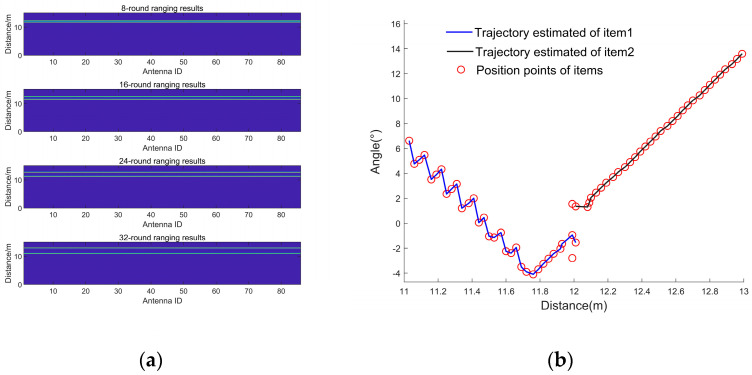
The results of distance domain compression and trajectory generation for a frame of the IF signal using the algorithm proposed in this paper. (**a**) Distance-domain compressed image of a frame with a partial period of an IF signal; (**b**) angular and positional points of the corresponding objects at different periods within a frame of the signal, generating trajectories by data correlation.

**Figure 20 sensors-23-04531-f020:**
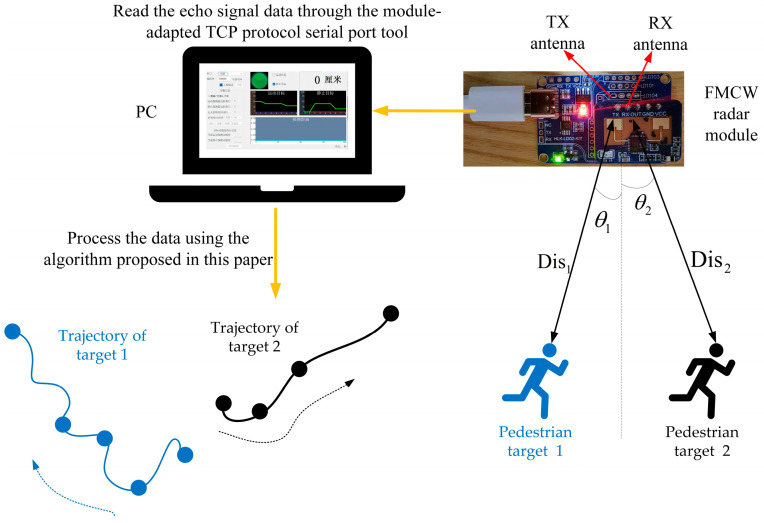
The experimental design for estimating the trajectory and velocity of an object using one frame of the IF signal.

**Figure 21 sensors-23-04531-f021:**
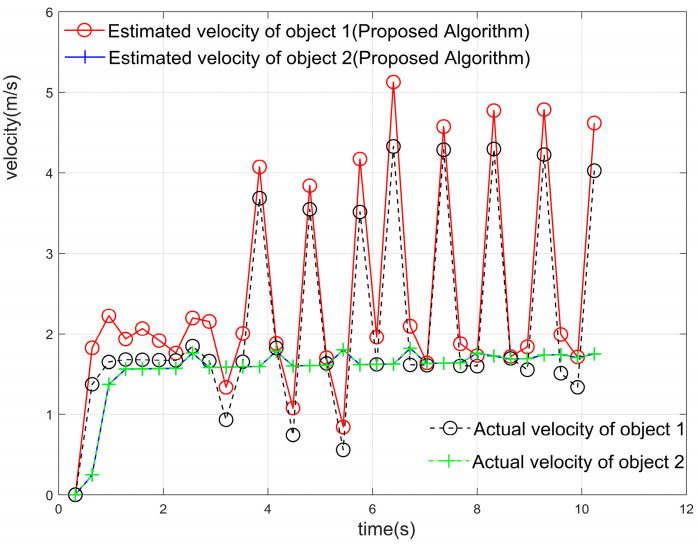
The estimation of the average speed of the single-frame signal of two objects using the algorithm proposed in this paper compared with the speed set in the simulation.

**Figure 22 sensors-23-04531-f022:**
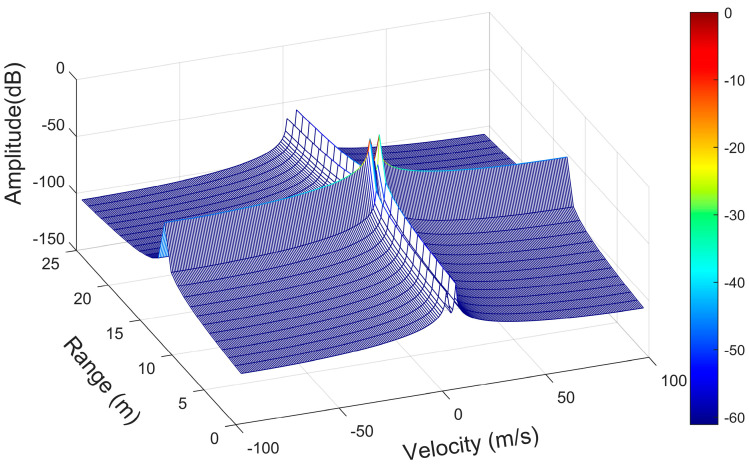
The three-dimensional distance-velocity domain FFT diagram of two objects in the 28th cycle of an IF signal frame.

**Figure 23 sensors-23-04531-f023:**
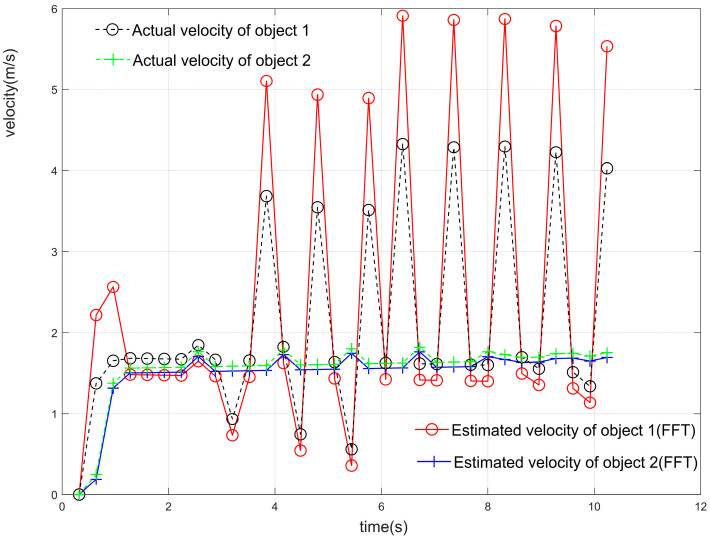
The results of velocity estimation of moving objects using velocity-FFT versus actual velocity.

**Figure 24 sensors-23-04531-f024:**
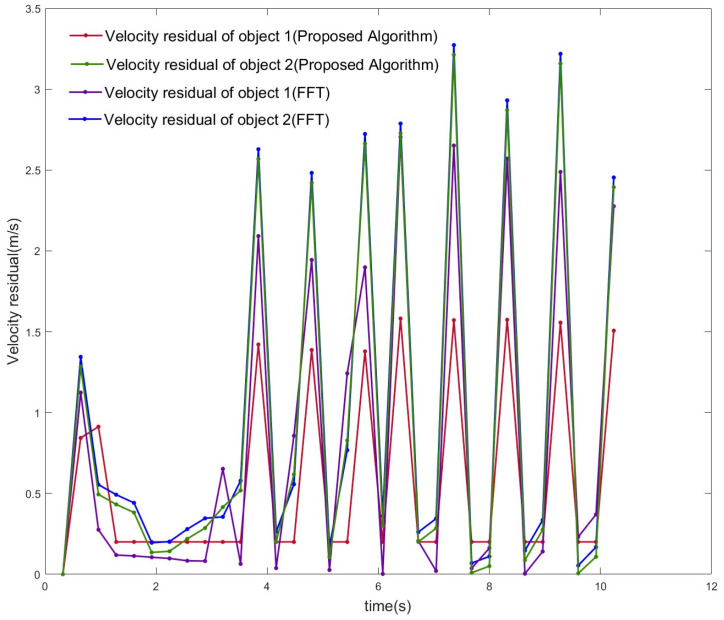
The comparison of the residuals of estimated velocity using two algorithms.

**Table 1 sensors-23-04531-t001:** The simulation parameters of the FM signal and the antenna array used in this paper are set as follows.

Simulation Experiment Parameter Type	Set Value
Sampling interval $T_{s}$	$1.25\times 10^{-7}\;s$
Chirp Periodicity T	$3.2\times 10^{-5}\;s$
Number of chirp signal periods in one frame of IF signal $N_{f}$	32
Aperture of the antenna array element *L*	0.0815 m
Number of antenna array elements $N_{a}$	12
Linear FM slope of the chirp signal emitted by the antenna γ	$78.986\times 10^{12}\;Hz/s$
Carrier frequency $f_{0}$	$78.8\times 10^{9}\;Hz$

**Table 2 sensors-23-04531-t002:** Angular resolution and runtime of a 24-array antenna using different algorithms at different SNRs.

	Different SNR	FFT	MUSIC	OMP/Algorithm Proposed	
Angle Resolution	20	19°	6°	0.10° ^1^	0.10°
	30	17°	4°	0.04°	0.05°
	50	15°	3°	0.04°	0.01°
Running Time	20	0.61 s	5.11 s	1.91 s ^2^	1.45 s
	30	0.59 s	5.08 s	1.84 s	1.37 s
	50	0.70 s	5.24 s	2.14 s	1.41 s

^1,2^ In the “OMP/Algorithm proposed” column of “Angle Resolution” and “Running Time”, the left column represents the angle resolution/running time of the traditional OMP algorithm, and the right column represents the angle resolution/running time of the algorithm proposed in this paper.

**Table 3 sensors-23-04531-t003:** Comparison of the positioning results of the antenna array obtained by particle swarm algorithm optimization with the original experimental settings.

	Setting Values for Five Experiments		Positioning Results of the Optimized Virtual Antenna Array Using the Algorithm Proposed in This Paper					
No.	Ang * (°)	Dis (m)	Ang Detected (°)	Ang Error (°)	Ang Error-to-Ang	Dis Detected (m)	Dis Error (m)	Dis Error-to-Dis
1	−0.30	11.99	−0.31	**0.01**	3.33%	11.99	0	0%
	0.30	12.21	0.35	0.05	16.67%	12.21	**0**	0%
2	−30.00	6.90	−29.80	0.20	0.67%	6.88	0.02	0.29%
	10.00	10.60	10.10	0.10	1%	10.61	0.01	0.09%
3	5.00	11.30	5.20	0.20	4%	11.39	0.09	0.80%
	40.00	8.60	39.91	0.09	0.23%	8.61	0.01	0.12%
4	−50.00	1.30	−49.10	**0.90**	1.8%	1.29	0.01	0.77%
	20.00	13.60	20.11	0.11	0.55%	13.78	**0.18**	1.32%
5	−10.00	9.30	−10.10	0.10	1%	9.29	0.01	0.11%
	4.00	7.60	3.96	0.04	1%	7.49	0.11	1.45%

* Ang means angle, and Dis means distance.

## Data Availability

The data for this paper comes from simulation settings and measurements during the experiment; please contact the corresponding author for additional data requests.

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
