# Peer review of "Research on a Super-Resolution and Low-Complexity Positioning Algorithm Using FMCW Radar Based on OMP and FFT in 2D Driving Scene"

_sensors, 2023, doi:10.3390/s23094531_

Round 1

Reviewer 1 Report

In this paper, the authors have presented a positioning algorithm to improve the accuracy of antenna aging. Experimental results have been presented and discussed.
The paper is quite well written and could be of interest to the community once some remarks will be addressed. Here are listed:

1) l.11 "FMCW" I suggest defining each acronym the first time you use it. Please define all the acronyms used in the paper.
2) l.80 caption fig. 1, I suggest rephrasing the caption with "This figure shows".
3) Fig.2 Please add a description of each element, what generator was used, what kind of antenna etc..
4) l.95 Why "2.5 GHz",  please add a comment.
5) Figg.3 What does B represent in eq.4? please add a comment.
6) l.131 please explain "dictionary"
7) l.147-148 how did you manage to measure +/- 50° and the error of 6°?
8) l.190 "AntDis", what is? please explain
9) In order to increase the readability of the paper, I suggest placing each pseudo-code at the end of the paper in the supplementary material section.
10) Fig. 13 Please add units in the vertical axis. Please do the same also in fig. 14.
11) Tab. 4, if I understood well, you are able to estimate 0.01°, and the angle measurement error was less than 10°. Was 10° overestimated?  can you add some comments?

Some minor changes should be addressed.

Author Response

Please see the attachment. We have finished our responses in attached word file.

Reviewer 2 Report

The paper proposed a PSO based algorithm to predict the angular and position information of targets from a FMCW radar. The algorithm itself tried to handle the situation when the antenna array for detection has an aging problem. In addition to the proposed algorithm, intermediate RF signals from the FMCW radar have also been processed to derive the motion trajectory based on range and velocity information of the object from each individual frame. Even though the algorithm and results are interesting to this reviewer, however, the structure of the paper is poorly organised, e.g., it included 22 figures. The sequence of presentation of the work seems not to follow a logic order. A lot of results poured together. Hence, this reviewer has the following suggestions to the authors in order to improve the quality of this paper:

1) Rewrite the abstract, clarifying what resolution was achieved, and how much complexity was reduced due to the proposed algorithm, and whether the same algorithm has been implemented in the motion trajectory estimation.

2) Reorganise the paper, make the OMP sparse decomposition in the section 2.1., put the super-resolution algorithm and motion trajectory algorithm in a separate first level section.

3) Fig. 1 is quite confusing to me, if you just try to point out the relations between different algorithms, there is a better way to make it clearer.

4) Equ. (1) and Equ. (2) has no connection.

5) Fig. 3 looks a bit odd, could you check why there are two intermediate frequencies.

6) How to construct Fig. 8 is not well explained, why the number of components in the angle domain is NTX x NRX?

7) The quality of Fig. 13 is poor. Information in the Fig. 14 is incomplete.

Language problems:

Such as this one:

This paper also further realizes the use of single frame intermediate frequency signal to estimate the position angle information of the object and obtain the motion trajectory and velocity”

Are you sure it is correct to say, "realizes the use of single frame intermediate frequency signal?" Probably you can just say: "In addition, we can use intermediate frequency signal within a single frame to estimate..."

There are lots of similar issues, please check it, such as: 

“The received signal reflected by the radar from an object at a distance of R0 is shown in equation 3”

I dont think the received signal is reflected by the radar.

Author Response

(The authors gave the same response as above.)

Reviewer 3 Report

The manuscript presents a low-complexity OMP-based super-resolution localization algorithm based on the frequency modulation continuous wave millimeter wave radar model for position and angle estimation of objects for two-dimensional scenarios of autonomous driving. This manuscript is well written and well organized. The results show good improvement and therefore I feel it can be accepted pending minor revisions:

1. All abbreviations should be described in the context, e.g. FMCW, OMP, SNR, etc. It is recommended to give a list of abbreviations at the end of manuscript.

2. Although the manuscript focuses on the frequency modulation continuous wave (FMCW) method, it is suggested to compare this method with other related methods such as phase shift method, at least in the literature survey.

3. The signal-to-noise ratio is important parameter which should be considered and properly addressed.

4. The selection of particle swarm algorithm (PSO) should be explained.

5. The text needs to be edited. For example Ref. [23] is mentioned before Ref. [5] and Figure 12 before Figure 8.

6. The quality of some figures is poor, for example Figs. 13, 16, and 18.

7. Adding the distance error-to-distance ratio and also the angle error-to-angle in Table 4 can be useful for readers.

Author Response

(The authors gave the same response as above.)
